# Interfacial Charge Transfer and Ultrafast Photonics Application of 2D Graphene/InSe Heterostructure

**DOI:** 10.3390/nano13010147

**Published:** 2022-12-28

**Authors:** Jialin Li, Lizhen Wang, Yuzhong Chen, Yujie Li, Haiming Zhu, Linjun Li, Limin Tong

**Affiliations:** 1State Key Laboratory of Modern Optical Instrumentation, College of Optical Science and Engineering, Zhejiang University, Hangzhou 310027, China; 2Center for Chemistry of High-Performance & Novel Materials, Department of Chemistry, Zhejiang University, Hangzhou 310027, China; 3Intelligent Optics & Photonics Research Center, Jiaxing Research Institute, Zhejiang University, Jiaxing 314000, China

**Keywords:** charge transfer, graphene/InSe heterostructure, pump–probe, nonlinear absorption, nonlinear photonic application

## Abstract

Interface interactions in 2D vertically stacked heterostructures play an important role in optoelectronic applications, and photodetectors based on graphene/InSe heterostructures show promising performance nowadays. However, nonlinear optical property studies based on the graphene/InSe heterostructure are insufficient. Here, we fabricated a graphene/InSe heterostructure by mechanical exfoliation and investigated the optically induced charge transfer between graphene/InSe heterostructures by taking photoluminescence and pump–probe measurements. The large built-in electric field at the interface was confirmed by Kelvin probe force microscopy. Furthermore, due to the efficient interfacial carrier transfer driven by the built-in electric potential (~286 meV) and broadband nonlinear absorption, the application of the graphene/InSe heterostructure in a mode-locked laser was realized. Our work not only provides a deeper understanding of the dipole orientation-related interface interactions on the photoexcited charge transfer of graphene/InSe heterostructures, but also enriches the saturable absorber family for ultrafast photonics application.

## 1. Introduction

Two-dimensional (2D) heterostructures, which are combined by van der Waals (vdWs) force, have attracted great attention for next-generation optoelectronic devices, because of their excellent physical properties, such as strong light–matter interactions and ultrafast interfacial charge transfer [1,2,3,4,5]. Graphene, as a well-known 2D material, has been widely used in optoelectronic areas due to its ultrafast electron relaxation time [6,7]. However, the main shortcoming of graphene for optoelectronic applications is the relatively low absorption [8,9]. Recently, an III–VI group layered semiconductor, InSe, also showed great potential in optoelectronics areas, with high carrier mobility [10], a tunable bandgap [10,11] and a high nonlinear absorption coefficient [12,13]. For example, InSe has been demonstrated as a broadband photodetector [14,15] and a saturable absorber (SA) in ultrafast fiber lasers [16,17,18,19] and solid-state bulk lasers [20]. Although photodetectors based on graphene/InSe vdWs heterostructures operating at the visible to near-infrared (NIR) wavelength range have been reported [21,22,23,24], they have been demonstrated to possess high photodetection performance for phototransistor applications due to the suitable band alignment and interfacial charge transfer; however, all the measurements presented in previous works were obtained under quasi-static conditions. To investigate the role of dipole orientation-related interface interactions in the dynamic photoexcited charge transfer process in graphene/InSe heterostructures, ultrafast pump–probe optical spectroscopy studies are required. In addition, 2D graphene/InSe-based heterostructures for nonlinear photonic applications have not been studied, but it is expected that the heterostructure combines both the advantages of ultrafast relaxation and a large effective nonlinear absorption coefficient for higher performance.

In this work, we prepared a graphene/InSe (G/InSe) heterostructure (HS) by mechanical exfoliation (ME) and investigated the intrinsic interlayer charge transfer process of G/InSe HS by steady-state photoluminescence (PL), Kelvin probe force microscopy (KPFM) and transient absorption (TA) pump–probe measurements. We further demonstrated the G/InSe HS as an SA for near-IR mode-locked laser application. Stable traditional soliton pulses were obtained with a central wavelength of 1566.65 nm and 192 fs pulse duration. These results indicate that G/InSe HS is very attractive for ultrafast nonlinear photonic applications.

## 2. Preparation and Characterization

Our Bridgman method-grown InSe crystals are obtained commercially (from SixCarbon Technology, Shenzhen, China). The XRD pattern of bulk InSe is shown in Figure 1a, demonstrating that the crystal structure is β phase, which belongs to the space group P63/mmc, and the lattice parameters are a = 4.05 Å, b = 4.05 Å, c = 16.93 Å, respectively. Raman scattering measurement on the InSe bulk single crystal is performed by a commercial Raman spectrometer (Witec alpha300, D-89081 Ulm, Germany) with a λ = 532 nm laser for excitation. As shown in Figure 1b, there exist three Raman vibration modes, namely A_1g_^1^ (115 cm^−1^), E_2g_^1^ (176 cm^−1^) and A_1g_^2^ (226 cm^−1^), similar to the results of ref. [25], which proves that the sample is β phase with high single-crystalline quality. Figure 1c shows a typical surface topographic image taken via scanning electron microscopy (SEM); the corresponding elemental ratio of InSe is obtained by energy-dispersive X-ray (EDX) spectroscopy, which reveals an In:Se ratio of 1.16, as shown in Figure 1d. The elemental mapping of In and Se is shown in Figure 1e,f, respectively, indicating the uniform elemental spatial distribution.

For the fabrication of the graphene/InSe heterostructure, the process can be divided into three steps. Firstly, a thin flake of graphene was prepared by the ME method using Nitto tape [26] and transferred to a quartz substrate. Secondly, a thin flake of InSe was exfoliated on PDMS [27] and then transferred to the top of the part of the graphene flake, assisted by a transfer station, under a Nikon microscopy. Thirdly, the sample was then thermally annealed in an Ar atmosphere under 200 °C for 2 h to remove chemical residues and improve the interfacial contact. Figure 2a shows an optical image of the G/InSe and its Raman spectrum is shown in Figure 2b. All the Raman peaks related to InSe and graphene (113 cm^−1^, 174 cm^−1^, 226 cm^−1^, 1581 cm^−1^ and 2716 cm^−1^) are observed in the heterostructure region, indicating the successful formation of the G/InSe heterostructure, and no Raman peaks are shifted compared with individual InSe and graphene. The PL measurements were also performed on a G/InSe sample derived by the same Raman spectrometer, using a 532 nm laser for excitation, with the power of 700 μW. As shown in Figure 2c, PL is quenched in the heterostructure region, indicating that the separation of photoexcited electron–hole pairs occurs at the G/InSe interface.

To determine the work function difference of graphene and InSe, Kelvin probe force microscopy (KPFM) was used. According to the equation V = (W_sample_ − W_tip_)/e, the difference in work function (W) represents the surface potential (V) variation of the sample. Figure 2d shows the KPFM curve of graphene and G/InSe. The result indicates that InSe is n-type doped, while graphene is heavily p-type doped, leading to the formation of a p–n junction with a large built-in electric potential (the potential difference is ~0.286 eV). Figure 2e shows the surface work function mapping image of graphene and G/InSe. Thus, electron transfer occurs from graphene to InSe, enhancing the p-type carrier concentration in graphene and the n-type concentration in InSe. The band alignment of the heterostructure is illustrated in Figure 2f according to the work function. The unique band alignment of G/InSe HS allows the possible transition from graphene to InSe under low photon-energy excitation, which can be used as an SA in C/L-band pulsed lasers. In addition, the large built-in electric field existing in the G/InSe heterostructure can inhibit electron–hole recombination and reduce the recombination rate, leading to a fast electron relaxation time, which is beneficial for achieving ultrafast laser pulses.

## 3. Carrier Dynamics

To probe the carrier transport process across the G/InSe interface and reveal the ultrafast nonlinear optical properties of the G/InSe in the visible–near-infrared (IR) spectral region, a micro-area pump–probe technique is employed. The TA measurement setup is shown in Figure 3a. Specifically, the output fundamental beam (λ = 1030 nm, ~170 fs pulse duration) is split into two paths from a Yb: KGW laser (Light Conversion Ltd. Pharos): one is sent into a noncollinear optical parametric amplifier (OPA) to produce a pump pulse at near ultraviolet, visible and near-IR wavelengths, and the other is focused onto a YAG crystal after a delay line to generate a probe pulse of white light continuum (λ = 500–950 nm) or near-IR (λ = 1425–1600 nm) light. The pump and probe beams are recombined and focused on the sample through a reflective 50× objective. The spot size of the focused femto laser is approximately 2 μm.

The transient absorption (TA) signal of InSe mainly comes from free carriers, and such carriers are the free holes of the valence band (VB1 and VB2) level. The photo-bleaching signature arising from Pauli blocking indicates the saturable absorption behavior of the prepared samples. Under excitation by a 2.06 eV, ~130 μJ cm^−2^ pump pulse, the TA dynamics of graphene, InSe and G/InSe HS are as shown in Figure 3b,c, in which the A exciton (844 nm) and B exciton (502 nm) of InSe are selected as the probe wavelength, respectively. Fitting their dynamics, the G/InSe HS shows a biexponential decay with an intra-band relaxation time (τ_1_ = 48 fs) and inter-band relaxation time (τ_2_ = 140 fs) that is closer to graphene (τ_1_ = 96 fs and τ_2_ = 583 fs) and much faster than InSe (τ_1_ = 1.77 ps and τ_2_ = 216.11 ps) at a probe wavelength of 844 nm, as seen in Figure 3a. The A exciton of InSe is associated with the transition from the principal bandgap, which has a distinct electric dipole-like character, coupled to out-of-plane polarized photons [28], so it has less absorption in our configuration, in which the laser polarization is in-plane (as shown in insert of Figure 3a). In contrast, the B exciton is related to the transition that is coupled to in-plane polarized light. At a probe wavelength of 502 nm, as seen in Figure 3c, the TA signal is attributed to two-photon absorption, and the relaxation time of the G/InSe HS is determined to be τ_1_ = 12.6 ps and τ_2_ = 12.6 ps, which is also faster than that of the InSe individual (τ_1_ = 3.8 ps and τ_2_ = 191.3 ps). Benefiting from graphene’s fast charge transfer and relaxation channel, the carrier recombination in InSe is suppressed, thus resulting in a short relaxation time. Furthermore, the TA dynamics of InSe and the G/InSe HS under the pump photon energy of 3.30 eV, ~2 μJ/cm^2^ are shown in Figure 3d. Compared to InSe, the relaxation time of the G/InSe HS is faster than the InSe itself, indicating the fast charge transfer at the interface of the G/InSe HS. In addition, the TA signal is stronger compared with the signal under 2.06 eV, ~130 μJ cm^−2^ at the probe wavelength of 502 nm (B exciton); therefore, it can be confirmed as two-photon absorption at the pump photon energy of 2.06 eV. Overall, the ultrafast carrier dynamics revealed the dipole orientation-related interface interactions in the G/InSe HS, and the results not only provide a deeper understanding of the dynamic photoexcited charge transfer process, but also suggest a high optical modulation speed for nonlinear photonics application.

## 4. Nonlinear Saturable Absorption and Mode-Locked Fiber Laser Applications

To measure the saturable absorption properties of the G/InSe HS, the micro I-scan with a balanced twin-detector measurement method is used, as illustrated in Figure 4a. The pulsed laser source is operated at 1550 nm with a 300 fs pulse duration and 100 kHz repetition rate. The sample is located at the focal point of the laser through a 50× objective lens, and an adjustable attenuation filter is used to modulate the laser power. As shown in Figure 4b, the obtained transmittance data are fitted by using the two-level saturable absorption model with the equation $\alpha =\alpha_{ns}+\alpha_{s}/\left(1+I/I_{s}\right)\;$ [29]. The saturable intensity and modulation depth obtained are 1.33 GW/cm^2^ and 12%, respectively. It should be noted that the large modulation depth and low saturation intensity are attributed to the lower recombination rate caused by the large built-in electric potential [30]. 

To evaluate its ultrafast nonlinear optical response, the G/InSe HS SA was inserted into a ring fiber cavity. Figure 4c shows the schematic of the all-fiber G/InSe HS ring laser. To pump a 0.4-m-long erbium-doped gain fiber (EDF; Liekki Er110-4/125, Camas, WA, USA), a commercial continuous-wave (CW) 980 nm laser diode (LD, Nozay, France) with maximum power of 800 mW was used as a pump source. To couple the pump laser into the cavity and prevent back reflection, ensuring unidirectional laser operation, a 980/1550 nm wavelength division multiplexer/isolator hybrid (WDM + ISO) was used. Two polarization controllers (PCs) were used to tune the laser polarization state in the cavity, and the cavity also comprised a 5.2-m-long single-mode fiber (SMF-28e). The dispersion parameters of EDF and SMF are 12 ps^2^/km and −23 ps^2^/km, respectively; therefore, the calculated net cavity dispersion is approximately −0.106 ps^2^. The laser was output through a 10% optical coupler (OC), and its characteristics were real-time-monitored by a 1 GHz photodetector (Thorlabs DET01CFC, Newton, NJ, USA) and then output to a 3-GHz digital oscilloscope (LeCroy WavePro7300, Chestnut Ridge, NY, USA). Moreover, a commercial optical spectrum analyzer (Yokogawa AQ6370D, Tokyo, Japan) was used to record the obtained laser optical spectrum. The laser pulse width was measured by a commercial autocorrelator (APE Pulsecheck-USB-50, Berlin, Germany), and the radio frequency spectrum was measured by a RF spectrum analyzer (Rigol DSA1030, Suzhou, China).

By carefully controlling the laser polarization state through PCs, self-starting mode locking is achieved when the pump power is beyond 40 mW, due to the saturable absorption of the G/InSe HS SA. The SA damage threshold is around 650 mW. The maximum output laser power is around 2.53 mW, corresponding to the single pulse energy of 0.06 nJ. The output pulse characteristics under the pump power of 110 mW are shown in Figure 4d,e. Figure 4c shows the typical output optical spectrum with several pairs of Kelly sidebands, which indicates the signature of conventional soliton operation. The spectrum shows a 3-dB bandwidth of 7.43 nm centered at 1566.65 nm. Figure 4e presents the typical output pulse trains. The time interval of mode-locked pulses is approximately 24.41 ns, well matched with the total cavity length. The RF spectrum around the fundamental repetition rate of 40.96 MHz with the signal-to-noise ratio of ~27 dB is shown in the inset of Figure 4e, which indicates the good stability of the G/InSe HS mode-locked laser. The G/InSe HS SA is stable over two weeks in an ambient environment, and the mode-locking operation can be stable for over 10 h. The output mode-locked pulses are amplified by a home-made erbium-doped fiber amplifier (EDFA) and compressed by a piece of dispersion-compensating fiber. As is illustrated in Figure 4f, the obtained mode-locking pulse duration is approximately 270 fs. By fitting the autocorrelation (AC) trace with a Gaussian function, the actual pulse duration is estimated to be approximately 192 fs.

Furthermore, we compared the mode-locked laser results with the naked graphene (before being stacked with InSe) SA, and it showed that the mode-locking pulse width was around 292 fs (Appendix A), slightly slower than that of the G/InSe HS, due to the slower electron relaxation time proved by the TA measurement at a probe wavelength of 1566 nm (Appendix A). The internal mechanism is the reduced recombination rate and fast interlayer electron transfer due to the large built-in electric field in the G/InSe HS. The thickness of thin graphene and InSe determined by atomic force microscopy (AFM) is 5 nm and 7 nm, respectively (Appendix A). Moreover, InSe shows a large effective nonlinear absorption coefficient (β_eff_ ~ −2.8 × 10^2^ cm/GW) [13], while the heterostructure possesses a larger nonlinear absorption coefficient [31]. Furthermore, the ultrafast electron transfer from graphene to other 2D semiconductors occurs from the visible to mid-infrared region [32]. Benefiting from the ultrafast relaxation time and strong broadband nonlinear absorption, we believe that the heterostructure could be realized for ultrafast broadband nonlinear optical applications, not only limited to the C/L band.

## 5. Conclusions

High-quality G/InSe HS material was prepared by ME and the dry-transfer method, and the carrier transport across the G/InSe HS interface was systematically investigated by PL, KPFM and TA measurements. The relatively lower saturation intensity (~1.33 GW/cm^2^) and larger modulation depth (~12%) were obtained by nonlinear absorption measurement. Furthermore, mode-locked laser pulses with 192 fs pulse duration operating at 1566.65 nm were successfully generated by integrating the G/InSe HS into an erbium-doped fiber laser cavity. The obtained pulse width was superior to that of the naked graphene SA in the same laser cavity. The internal mechanism was the reduced rate of recombination and fast interlayer charge transfer driven by the large built-in electric potential of the G/InSe HS. Our work not only provides a deeper understanding of the dipole orientation-related interface interactions of the G/InSe HS, but also demonstrates its application prospects in the field of nonlinear photonics.

## Figures and Tables

**Figure 1 nanomaterials-13-00147-f001:**
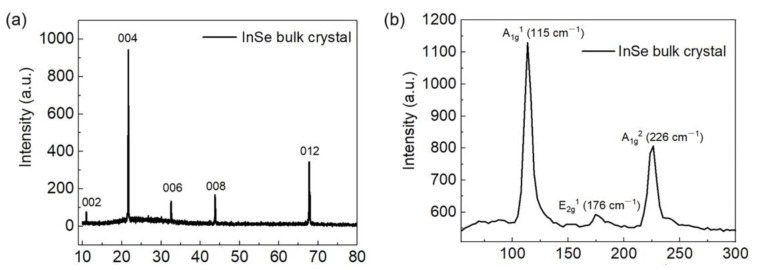
(**a**) XRD spectrum of InSe bulk crystal. (**b**) Raman spectrum of InSe bulk crystal. (**c**) SEM image of InSe bulk crystal. (**d**) EDX of InSe bulk crystal. (**e**,**f**) Elemental mapping image of InSe bulk crystal.

**Figure 2 nanomaterials-13-00147-f002:**
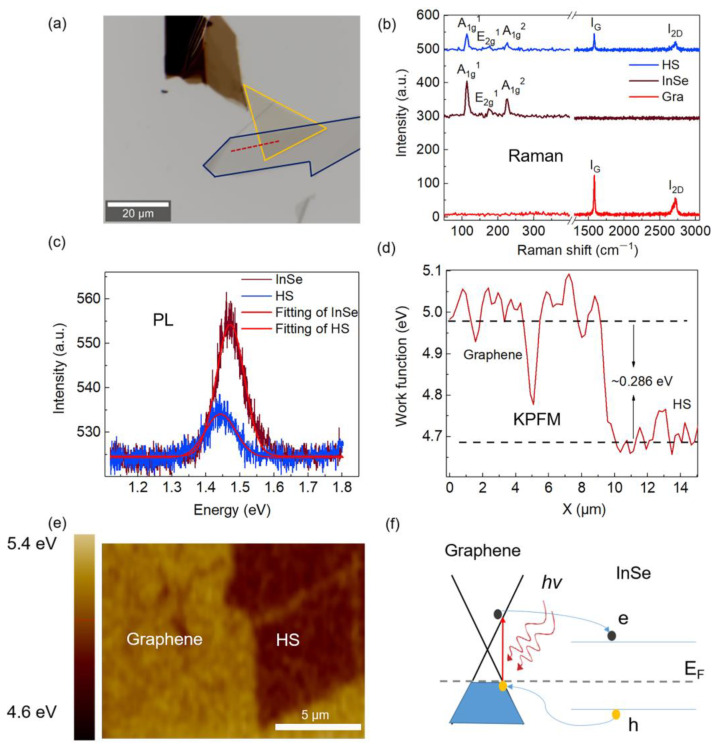
(**a**) Optical image (the blue line is bottom graphene and the yellow line is top InSe), (**b**) Raman spectrum, (**c**) PL spectrum of graphene, InSe and G/InSe HS. (**d**) Work function difference of graphene and HS (corresponding to the red dashed line region in (**a**). (**e**) KPFM surface work function image of graphene and HS. (**f**) The band alignment of the heterostructure.

**Figure 3 nanomaterials-13-00147-f003:**
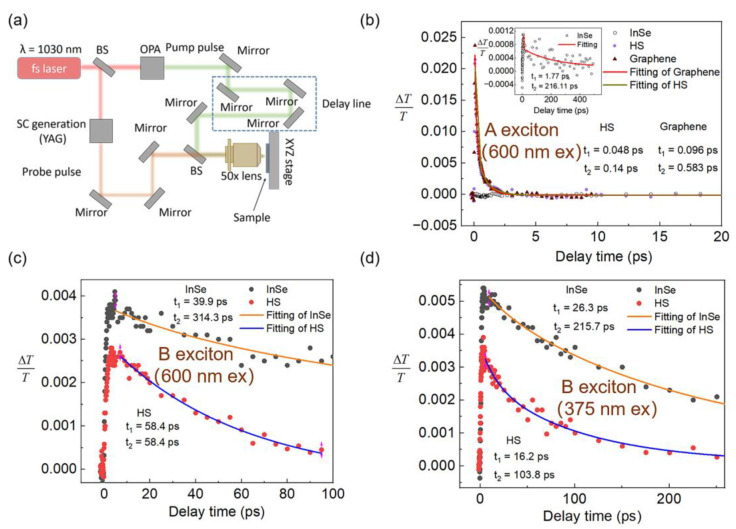
(**a**) Schematic diagram of the micro-area TA setup. (**b**) TA dynamics of graphene, InSe and G/InSe HS with pump and probe wavelengths of 600 nm and 844 nm (A exciton), respectively. (**c**) TA dynamics of InSe and G/InSe HS with pump and probe wavelengths of 600 nm and 502 nm (B exciton), respectively. (**d**) TA dynamics of InSe and G/InSe HS with pump and probe wavelengths of 375 nm and 502 nm (B exciton), respectively.

**Figure 4 nanomaterials-13-00147-f004:**
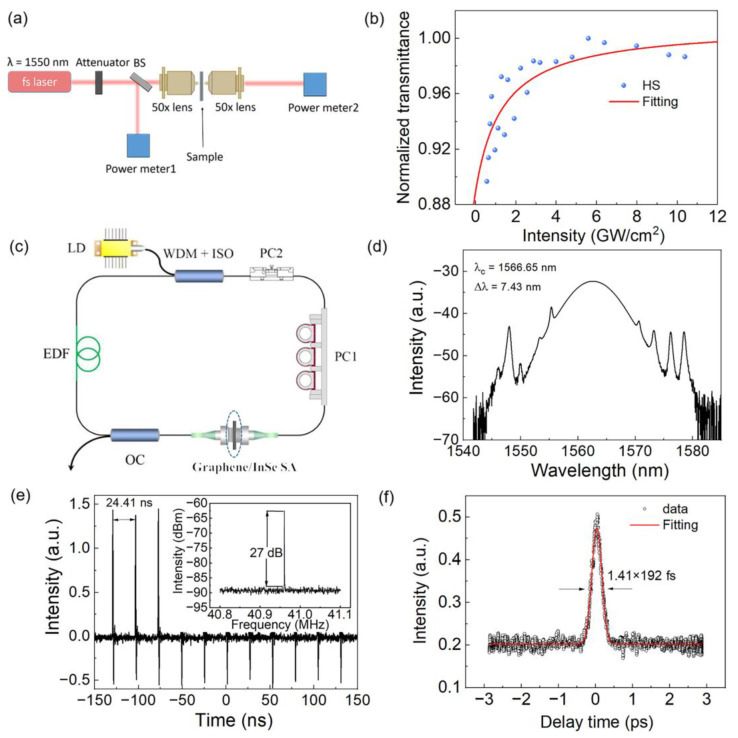
(**a**) Schematic diagram of the micro I-scan measurement system. (**b**) Nonlinear saturable absorption curve of G/InSe HS. (**c**) Schematics of the mode-locked fiber laser based on G/InSe HS SA. (**d**) Output optical spectrum based on G/InSe HS SA (the spectral resolution is 0.02 nm). (**e**) Mode-locked pulse trains. Inset figure is the obtained RF spectrum with 100 Hz resolution bandwidth. (**f**) AC trace of mode-locked pulses after amplification and compression.

## Data Availability

The data can be requested from the corresponding author upon reasonable request.

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
