# Peer review of "Interfacial Charge Transfer and Ultrafast Photonics Application of 2D Graphene/InSe Heterostructure"

_nanomaterials, 2022, doi:10.3390/nano13010147_

Round 1

Reviewer 1 Report

The author's report on an interfacial charge transfer and ultrafast photonics application of two-dimensional (2D) graphene/InSe heterostructure fabricated via mechanical exfoliation. The optically induced charge transfer for the heterostructures is investigated using photoluminescence studies and pump probe measurements. The built-in electric field at the interface of graphene and InSe is confirmed by Kevin probe force microscopy. The efficient interfacial carrier transfer and broadband nonlinear absorption paved way for the realization of mode-locked laser. This work contributes significantly towards the potential ultrafast photonics applications.

The authors present an incredibly good concise and informative introduction with references to the relevant and most recent literature. The articles contain important and significant novel results that will be of great interest for the broad scientific readership of the journal.

For the reasons mentioned above, I suggest this work deserves publication in Nanomaterials, provided the following (major) comments and suggestions are fulfilled:

1.       The authors have used SA abbreviation in the abstract and throughout the manuscript. The authors must define SA.

2.       The table in Figure 1b overlaps or hides the EDX graphical spectra. The authors must make the changes accordingly for better readership.

3.       The authors have provided Raman spectrum, SEM and EDX of InSe bulk crystal. Additionally, the authors should consider providing the crystal structure, lattice parameters and X-Ray diffraction pattern for the bulk InSe crystal.

4.       The Figure 2b shows the Raman spectrum of graphene, InSe and the heterostructure. The authors must define the peaks and discuss the significance of each corresponding to graphene, InSe, and the heterostructure.

5.       The authors should discuss the number of layers of graphene and InSe individually and in the heterostructure. Consider using I2D/IG and ID/IG ratio to explain the number of layers and quality of graphene obtained. Comment on the dependence of transient absorption on the number of layers of graphene and InSe in the heterostructure.

6.       The authors should provide atomic force microscopy image and height profile of the exfoliated graphene and InSe.

Reviewer 2 Report

Manuscript: “Title: Interfacial Charge Transfer and Ultrafast Photonics Application of 2D
Graphene/InSe Heterostructure”

Journal: Nanomaterials

The manuscript reports a study of  dynamical photoexcited charge transfer process in Graphene/InSe heterostructure, mechanical exfoliated, by PL, Kelvin probe, transient absorption and pump-probe measurements.

Although the work is interesting, and well written on average, it still requires attention on some points before being accepted in Nanomaterials.

Below are the comments and suggestions that must be sorted out before consideration for publication.

- Line 5: (in the abstract), The sentence: “ However….” is a bit ambiguous.
Please, clarify.

- Line 24: SA is not defined.

- The false color of Figure 1c and 1d are very difficult to read. Improve these panels.

-Eliminate the word: “obvious” from line 62.

- Line 68: Figure 2c and 2d is indeed 1c and 1d.

- line 73: Add a sentence that better describe the preparation of samples, Ref- [25]

-line 75: Add a sentence that, also here details the sample preparation Ref. [26].

- Figure 2 is too dark.

- Figure 2b is unreadable:separate the spectra vertically, and indicate the Raman transitions.

-Lines 78-83 Specify better how/where the PL measurements were performed.

- What "Individual InSe" does mean? Specify the sample thicknesses (for both InSe and graphene flakes).

- I would like to understand if the G/InSe heterostructure is,for both structure, at single layer. (Discuss this point at the light of Raman features).

- Improve the discussion of results reported in Figure 2 and their interpretation.

- Line 81: Why: “rapid”? At this point the word rapid is inappropriate.

- Why: “strong” electronic coupling? Graphene and InSe probe VdW adhesion, moderate/eliminate the word: “strong”.
(I suggest to improve also this part).

- The yellow triangle and losing in Figure 2a are not mentioned.

- Describe better this “dynamical” band alignment of Figure 2d, and the HS PL intensity reduction and shift of Figure 2c.

Please, revise the English.

In its present form the manuscript cannot be accepted in Nanomaterials (MDPI), and requires major revision.

Round 2

Reviewer 1 Report

The authors have adequately addressed the reviewers' comments. The manuscript can now be published in the present form. 

Reviewer 2 Report

In my opinion, the manuscript is now ready to be considered in Nanomaterials, having authors taken into account referee's comments/suggestions.